# POPULATION DESCENT HYPER-PARAMETER TUNING

## ABSTRACT

Gradient descent is currently the algorithm of choice for optimizing neural networks. However, much of its performance is dependent on the chosen hyperparameters for the optimizer as well as the model (e.g., learning rate, regularization rate, etc...). Traditional hyperparameter tuning techniques are computationally costly and disregard a model's current progress, diminishing their effectiveness. With these limitations in mind, we therefore present the first memetic algorithm specifically designed to tune gradient descent's hyperparameters using a population-based evolution scheme. PopDescent leverages an $m$-elitist selection strategy and a normalized fitness-based randomization scheme, both of which facilitate more sophisticated hyperparameter optimization. Our experiments demonstrate that PopDescent surpasses existing hyperparameter tuning methods, boosting performance by as much as 13%. We also demonstrate PopDescent's insensitivity to changes to its own hyperparameters, a key requirement for hyperparameter tuning not satisfied by state-of-the-art memetic algorithms.

## 1 INTRODUCTION

Today's machine learning methods almost entirely rely on gradient descent as a core optimization technique. Many recent deep learning tasks, whether it is supervised learning, or unsupervised, include Neural Network (NN) optimization with ever-growing parameter spaces. However, these methods have multiple hyperparameters (e.g., learning rate, regularization rate, batch size, model weights, etc...), all of which may affect training results significantly Dauphin et al. (2014); Brea et al. (2019). Furthermore, the set of optimal hyperparameters depends on the task at hand, and manually searching for them is both time-consuming and computationally expensive. Thus, the creation of hyperparameter tuning frameworks is an extensive research topic, and numerous hyperparameter tuning and meta-learning methods have been proposed to solve this problem Akiba et al. (2019); Rogachev & Melikhova (2020). Meta-learning methods focus on finding generalized models allowing faster convergence when trained across similar tasks, instead of optimal hyperparameter selection for each problem, making meta-learning techniques tangential to this paper Patacchiola et al. (2023).

Existing hyperparameter-tuning literature falls into three main categories. 1) They fully train models, evaluating the final results on a single hyperparameter combination; this is computationally wasteful Liashchynskyi & Liashchynskyi (2019) 2) They select hyperparameters based on the first few training epochs (making them ineffectual choices later on during training) Rogachev & Melikhova (2020). 3) They adjust hyperparameters based on a predetermined schedule Wu et al. (2019). One important limitation shared across these methods is the lack of "active" hyperparameter optimization, meaning they adjust hyperparameters without taking into account the model's current progress throughout the duration of training. Thus, they have no knowledge of the current loss space, and have no means of exploring the loss function outside of the model's current gradient.

Instead of dealing with fine-tuning hyperparameters, global optimization methods, usually population-based, employ non-differentiable optimization. Evolutionary/genetic algorithms Bäck (1996); Liashchynskyi & Liashchynskyi (2019) are one of the most popular methods that utilize mutations; differential evolution Storn & Price (1995) is a subset that uses differentiable mutations for faster convergence Karaboga & Cetinkaya (2005). Still, evolutionary algorithms suffer the curse of dimensionality, and are often not comparable to fine-tuned gradient-based solutions in terms of efficiency and performance Bhattacharya (2013).

Memetic algorithms combine evolutionary algorithms with a local search process (usually gradient descent) D'Angelo & Palmieri (2021); Moscato (1999); Xue et al. (2021). They bridge the gap between exploring more space based on random noise (evolutionary step), and efficiently exploiting the gradient (local step). *Memetic algorithms are not currently utilized for hyperparameter search*, but rather are used for specialized optimization problems.

Thus, we propose Population Descent (POPDESCENT), the first memetic algorithm purposed for local hyperparameter optimization. The key idea in POPDESCENT is *actively* choosing how much to explore the parameter and hyperparameter space or exploit a specific location on the loss curve. This decision is taken based on a normalized fitness score representing each model's progress after each iteration of local updates, and an m-elitist selection process to choose which models to keep, and which models to mutate. Inspired by using global population-based evolution to tune local search hyperparameters, we add a constantly adaptive learning and regularization rate for each iteration of gradient steps, helping accelerate or decelerate learning and generalization of each NN based on current progress. Non-differential based mutations for these rates allow for simple randomization by adding Gaussian noise; we also add Gaussian noise to NN weights.

We show in our evaluations that our memetic framework, uniquely designed to tune local search hyperparameters, yields better results compared to existing hyperparameter tuning methods such as grid search, while also outperforming state-of-the-art memetic algorithms such as ESGD Cui et al. (2018). To demonstrate POPDESCENT's ability to more effectively traverse the loss landscape on real deep-learning workloads, we apply the algorithm to the FMNIST, CIFAR-10, and CIFAR-100 classification benchmarks Xiao et al. (2017); Krizhevsky et al. (2009). We also conduct ablation studies justifying the effectiveness of the choices made in POPDESCENT. POPDESCENT achieves better test and training loss on every experiment we performed, while taking a lower number of total gradient steps (making it more efficient as well). We claim the following contributions:

- The first memetic algorithm specifically designed to tune local search hyperparameters

- A more *active* tuning framework compared to existing tuning methods through the use of an evolutionary step: learning/regularization rates, and NN weights are proportionally perturbed by Gaussian noise after *each local search step for the duration of training*

- A simplified memetic framework that is not sensitive to its own hyperparameters, a key requirement for hyperparameter tuning

- An open source reference implementation based on TensorFlow 2 which can be used directly in machine learning tasks.

## 2 POPULATION DESCENT

POPDESCENT is a memetic algorithm, meaning it combines both meta-optimization and gradient-based optimization into a single scheme. We define the pseudocode of POPDESCENT in Algorithm 1 which we hereafter describe in detail.

### 2.1 ALGORITHM DEFINITION

The goal of POPDESCENT is to find an optimal set of parameters forming what we term an individual. An individual is constructed from sets of parameters $\theta$, and hyperparameters $\alpha$. We search for individuals which maximize a user-provided FITNESS function. These individuals are maximized on batches from a held-out **Test** distribution that remains unseen during the procedure. Namely

$$\text{individual}^* = \langle \theta^*, \alpha^* \rangle = \sup_{\langle \theta, \alpha \rangle \in \textbf{Individuals}} \mathbb{E}_{\text{batch} \sim \textit{Test}} \left[ \text{FITNESS}(\langle \theta, \alpha \rangle, \text{batch}) \right]$$

However, since the **Test** distribution must remain unseen, we are forced to make use of available proxy data in the form of a **Training** distribution and a **CrossValidation** distribution. This is standard in machine learning workloads. We do not make assumptions on the differentiability of the provided FITNESS function. This allows one to use of common metrics of performance such as accuracy. Since the dimensionality of the parameter space can be exceedingly large (such as with Neural Networks), we make use of a LOCAL UPDATE function which can efficiently update the bulk of the parameters held in the $\theta$ of every individual. We assume that invocations of LOCAL UPDATE maximizes the individual's expected FITNESS over the **Training** set. An example of such a function is Stochastic Gradient Descent (SGD) as defined in Algorithm 2. SGD makes use of gradient-backpropagation to update $\theta$ in linear time. LOCAL UPDATE minimizes a differentiable LOSS as a proxy for maximizing FITNESS with respect to $\theta$. However, the LOCAL UPDATE function does not

modify the $\alpha$ hyperparameters. This can for example be the learning rate in SGD, and it can also be the regularization magnitude.

In order to find the best hyperparameters, POPDESCENT takes an $m$-elitist approach by holding onto a candidate set of individuals called a **Population**. In each iteration, The $m$ fittest individuals from the **Population** are kept untouched ($m$-elite), while the weakest ($|\textbf{Population}| - m$) individuals are always replaced. We then pick replacements from the **Population** but bias our choices towards fitter individuals. These replacements then go through a MUTATE operation provided by the user. The mutation magnitude depends on the fitness of the individual. That is, we mutate individuals more when they perform poorly. In a sense, the normalized FITNESS value allows the algorithm to be aware of progress made during optimization, and explore more space when that is more beneficial. Throughout the algorithm, $|\textbf{Population}|$ remains an invariant.

---

**Algorithm 1** POPDESCENT

---

**Require:** individual : $\theta \times \alpha$
**Require:** FITNESS : individual $\times$ **Batch** $\rightarrow [0, 1]$
**Require:** CONVERGED : individual $\rightarrow \{0, 1\}$
**Require:** LOCAL UPDATE : individual $\times$ **Batch** $\rightarrow$ individual
**Require:** MUTATE : individual $\times [0, 1] \rightarrow$ individual
**Require:** **Training** : $Distr[\textbf{Batch}]$
**Require:** **CrossValidation** : $Distr[\textbf{Batch}]$
**Require:** **Population** : $\{\text{individual}, \ldots\}$
**Require:** $m : \mathbb{N}$

1: **while** $\neg$ CONVERGED(**Population**) **do**
2:     batch$_{\textbf{Training}} \sim$ **Training**
3:     **Optimized** $\leftarrow \{$LOCAL UPDATE(individual, batch$_{\textbf{Training}}$) $\mid$ individual $\in$ **Population**$\}$
4:     batch$_{\textbf{CV}} \sim$ **CrossValidation**
5:     FITNESS$_{\textbf{CV}}(x) =$ FITNESS$(x, \text{batch}_{\textbf{CV}})$
6:     **WeightedMultinomial** $\leftarrow Pr(X = x) = \begin{cases} x \in \textbf{Optimized} & \frac{\text{FITNESS}_{\textbf{CV}}(x)}{\sum_{o \in \textbf{Optimized}} \text{FITNESS}_{\textbf{CV}}(o)} \\ x \notin \textbf{Optimized} & 0 \end{cases}$
7:     **Mutated** $\leftarrow \emptyset$
8:     **Strong** $\leftarrow$ **Optimized**
9:     **for** $1 \ldots (|\textbf{Population}| - m)$ **do**
10:        weak $\leftarrow \underset{\text{FITNESS}_{\textbf{CV}}}{\text{MINIMUM}}(\textbf{Strong})$
11:        **Strong** $\leftarrow$ **Strong**$/\{\text{weak}\}$
12:        replacement $\sim$ **WeightedMultinomial**
13:        **Mutated** $\leftarrow$ **Mutated** $\cup \{$MUTATE(replacement, $1 -$ FITNESS$_{\textbf{CV}}$(replacement))$\}$
14:     **end for**
15:     **Population** $\leftarrow$ **Strong** $\cup$ **Mutated**
16: **end while**
17: **return** $\underset{\text{FITNESS}_{\textbf{CV}}}{\text{MAXIMUM}}(\textbf{Population})$

---

POPDESCENT terminates when the user-defined CONVERGED function outputs 1 (line 1). Then, at each iteration:

1. Lines 2-3: The individuals in the **Population** all take a LOCAL UPDATE step over a batch sampled form the **Training** distribution. This produces a set of **Optimized** individuals;

2. Lines 4-5: A batch is sampled from the $CV$ distribution, upon which we build FITNESS$_{\textbf{CV}}$, i.e., the fitness function for that batch;

3. Line 6: We use FITNESS$_{\textbf{CV}}$ to build a **WeightedMultinomial** probability distribution whose samples are individuals from the **Optimized** set. The probability of each individual is defined by normalizing their fitness values, so that the probabilities sum to 1. This distribution is biased towards choosing fitter replacements;

4. Line 7-14: We iterate ($|\textbf{Population}| - m$) times replacing the ($|\textbf{Population}| - m$) lowest fitness individuals by a mutated replacement. We find replacement individuals via sampling from the ***WeightedMultinomial*** distribution (Line 12). Then the replacement is mutated by an amount dependent on its fitness: the lower the fitness, the more it will be mutated;

5. Line 15: **Population** is now updated to include the $m$ **Strong** individuals and the ($|\textbf{Population}| - m$) mutated replacements;

6. Line 17: Finally, we return the individual in the **Population** with the largest fitness.

---

**Algorithm 2** Example function implementations

---

**Require:** LOSS : individual $\times$ **Batch** $\to \mathbb{R}$
**Require:** $\beta_1, \beta_2 : \mathbb{R}$

    **function** LOCAL UPDATE(individual, **Batch**)
        optimized $\leftarrow$ individual
        optimized$_\theta \leftarrow$ individual$_\theta$ + individual$_\alpha \nabla_{\text{individual}_\theta}$ LOSS(individual, **Batch**)
        **return** optimized
    **end function**

    **function** MUTATE(individual, magnitude)
        mutated $\leftarrow$ individual
        mutated$_\theta \sim$ ***Gaussian***(individual$_\theta$, $\beta_1$magnitude)
        mutated$_\alpha \sim$ ***LogNormal***(individual$_\alpha$, $\beta_2$magnitude)
        **return** mutated
    **end function**

---

In the example function implementations in Algorithm 2, we also show a sample MUTATE function where we randomize the $\theta$ parameters via a ***Gaussian*** distribution whose standard deviation is defined by the mutation magnitude. We opt to modify the learning rate geometrically via a ***LogNormal*** distribution so that multiplying the learning rate by 0.1 and 10 is equally as likely with a standard deviation of 1. Note that when the magnitude is at 0, none of the parameters would change.

### 2.2 KEY POINTS IN POPDESCENT'S DESIGN

- We designed POPDESCENT to naturally select individuals which generalize well to a dataset not seen during local updates. We hypothesize that this would allow proper selection of regularization values rather than using ad-hoc techniques such as early stopping. This is evaluated in Section 3.3.

- If we remove the selection and mutation procedure then POPDESCENT simply becomes the random hyperparameter search algorithm, since after initialization, the individuals will be undergoing only iterations of SGD.

- POPDESCENT is also amenable to parallelization and the only synchronization required occurs in the replacements step.

- POPDESCENT has a few hyperparameters itself (depending on the implementation of MUTATE), but we have left these values constant across our experiments to showcase the effectiveness of the method and its low sensitivity to specific values of these parameters.

### 2.3 LIMITATIONS

Due to the no free lunch theorem Wolpert & Macready (1997), there will always be a case where this algorithm will be worse than a purely random approach at maximizing our FITNESS. For example, if the learning rate is initialized too high, too many randomization steps would be needed for making progress, due to the random walk nature of the mutation method used. Another limitation is that the algorithm does not take into account the best individual ever observed, meaning there is no guarantee that the algorithm will always improve in performance with more iterations. This is due to the decision to always take a LOCAL UPDATE with respect to the **Population**.

## 3 EVALUATIONS

In this section, we demonstrate that **1)** POPDESCENT's active tuning framework achieves better performance than existing tuning and memetic algorithms on the FMNIST, CIFAR-10, nd CIFAR-100 benchmarks; **2)** While significantly simpler, POPDESCENT converges at rates similar to the state-of-the-art memetic algorithm in a fair comparison; **3)** POPDESCENT's specific randomization scheme contributes to its results; and **4)** POPDESCENT is notably insensitive to changes in its own hyperparameters, allowing it to tune target parameters without having to tune the framework itself.

### 3.1 BENCHMARKS

We compare POPDESCENT against 1) grid search, due to its prevalence, 2) KerasTuner (KT): RandomSearch, due to its popularity (KT Rogachev & Melikhova (2020), and 3) Evolutionary stochastic gradient descent (ESGD) Cui et al. (2018), as it is the state-of-the-art memetic algorithm for benchmark machine learning workloads.

For clarification, KT's RandomSearch algorithm does not just randomly sample a subset of hyperparameters that would be explored during a grid search. Sampling is not limited to discrete rates (i.e. it can choose from continuous distributions), differentiating it from grid search. Also, RandomSearch chooses the "best" hyperparameters after testing parameter combinations on the first few (in our case, two) epochs of training, then resetting the model again, seeing which combination has the best validation loss early on. Thus, it can test more parameter combinations in fewer gradient steps.

*Some notes on benchmarks.* For the FMNIST and CIFAR-10 benchmarks, we opted to train models (4,575,242 and 186,250 parameters respectively) that are capable of overfitting. This makes these problems well-suited to evaluate these tuning frameworks. The "With Regularization" models use the same model with l2 kernel regularization added. To compare fairly against the available implementation of ESGD which does not implement regularization, we exclude comparisons with regularization. All algorithms use cross-validation loss as the metric for evaluating model fitness during training. All algorithms use the Adam optimizer for local search, except for ESGD, which uses SGD. Grid search "Without Regularization" trains five models each with a unique learning rate ([0.01, 0.001, 0.0001, 0.00001, 0.000001]). For "With Regularization," we let grid search enumerate the cartesian product of the five aforementioned learning rates and five different regularization rates producing 25 trained models. We use KT RandomSearch with 25 trials (# of combinations it tests). It samples learning rates from the continuous range of $[0.01 - 0.0001]$, and regularization rates from $[0.1 - 0.000001]$. RandomSearch and ESGD train over the whole dataset, and POPDESCENT and grid search sample portions of the data. A *gradient step* is defined by a single step taken by a local optimizer over one batch. We calculate the total number of gradients steps taken by every algorithm via total $=$ iterations $\times$ number of models $\times$ batches. We choose how many gradient steps to run each algorithm by observing when they converge. Our objective is minimizing the final test loss.

**FMNIST Benchmark.** We tested each algorithm on the Fashion MNIST image classification dataset in Figure 1, containing a 60k image training-set and a 10K image test-set (we split the test-set into two halves for a validation-set for all methods except ESGD, which uses the full test-set for validation). Each image is size 28x28, with 1 color channel. An identical Convolutional NN was used for each test (4,575,242 parameters), with three convolutional layers and two fully connected. Batch size is set to 64, and ESGD/POPDESCENT are initialized with learning rates of 0.001.

**CIFAR-10/100 Benchmark.** We tested each algorithm on the CIFAR-10 and CIFAR-100 image classification dataset in Figure 1, containing a 50k image training-set and a 10K image test-set, splitting test/validation loss exactly the same as for FMNIST. Each image is size 32x32 with 3 color channels. An identical Convolutional Neural Net was used for each test (186,250 parameters), with four convolutional layers and two fully connected. Batch size is set to 64, except for ESGD where we set it to 8, as is done in Masters & Luschi (2018). ESGD's learning rate is initialized to 0.01, and POPDESCENT to 0.001 (we found ESGD performs much better with 0.01 over 0.001).

**Benchmark Results.** POPDESCENT finds models with the lowest overall test loss across the board. It is also always taking the fewest or near fewest gradient steps. Both grid search and RandomSearch cannot adjust their parameters on-line, thus their convergence rates suffer. ESGD is our closest comparison to POPDESCENT as a memetic algorithm, but does not tune any hyperparameters. These results show ESGD's mutations perform well, but ESGD relies on either a static learning rate or

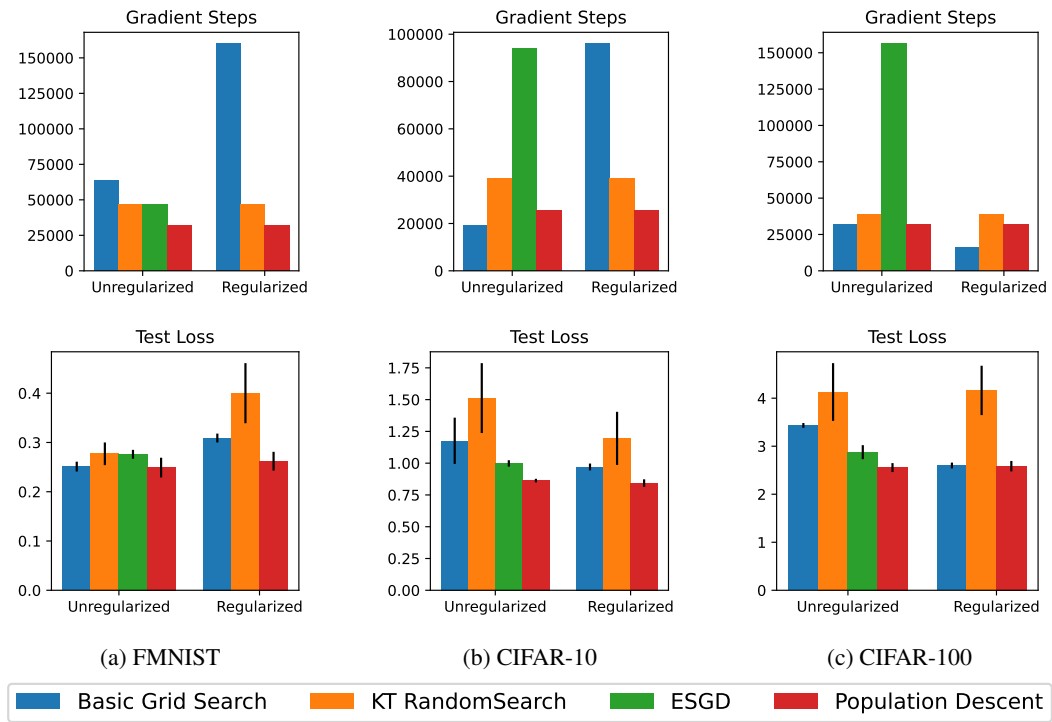

Figure 1: Benchmark Tests

a schedule, both of which struggle to keep up with POPDESCENT. On models that are capable of overfitting, POPDESCENT's ability to constantly monitor a model's performance on the cross-validation set and accelerate or decelerate its learning/regularization proves to be performant.

### 3.2 CONVERGENCE

Memetic algorithms like ESGD often rely on mutation lenghts, reproductive factors, mixing numbers, etc.; their genetic schemes are complex, and thus difficult to implement.

On the other hand, POPDESCENT's mutation step only adds independent noise to parameters, and uses a simple rank-based ($m$-elitist) recombination step. Still, when comparing convergence of the highest fitness model in the population, Figure 2 shows POPDESCENT converges to a lower validation loss faster than existing tuning methods and memetic algorithms.

We train each algorithm on six random seeds, running them for more iterations than optimal to show convergence/divergence over time (Grid Search: 100 iterations, KT RandomSearch: 25, POPDESCENT 115, and ESGD: 15). We plot the mean exponential moving average (bold line) of the cross-validation loss of the best model for each algorithm across all seeds, with the standard deviation (shading) for each algorithm's trials, as a function of gradient steps taken.

In Figure 2, RandomSearch is flat until about 46K gradients steps because it takes gradient steps to test which parameters are best without actually training the final model; it only trains the model after 25 trails (46K steps). Grid search and RandomSearch both overfit and struggle to reach a low loss due to non-dynamic tuning. POPDESCENT and ESGD are most succesful during training, though POPDESCENT achieves better final test loss with lower standard deviation, and requires fewer tunable hyperparameters to implement its global step.

### 3.3 ABLATION STUDY

This section analyzes how 1) the *randomization scheme* of NN weights/learning rate/regularization rate, and 2) the use of *cross-validation loss to evaluate the fitness* of individuals affects POPDES-

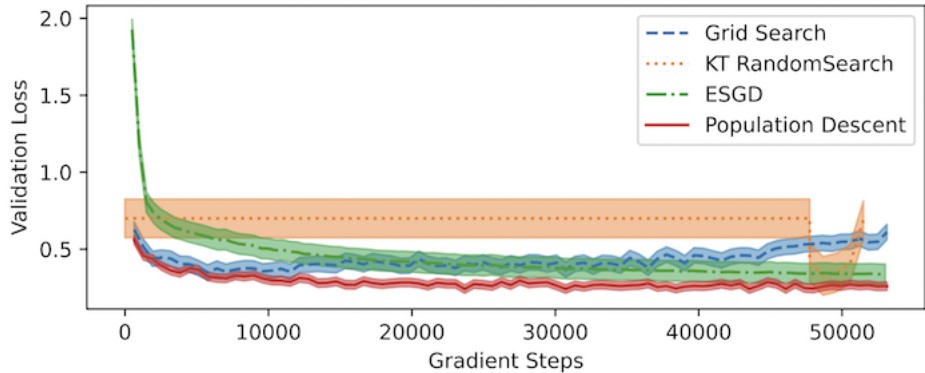

Figure 2: FMNIST validation loss progress.

CENT's performance. To emphasize the differences, we add l2 kernel regularization to every layer in the benchmark FMNIST model, and reduced the training set size to 10K. All tests are initialized with a default learning and regularization rate of 0.001. We choose $|\textbf{Population}| = 10$ and $m = 5$.

Table 1: Ablation study FMNIST

| Randomization | CV Selection | Regularization | Test Loss $\pm \sigma$ | Train Loss $\pm \sigma$ |
|:---:|:---:|:---:|:---:|:---:|
| **Ablation Study Over POPDESCENT Randomization** | | | | |
| ✓ | ✓ | ✓ | $0.345 \pm 0.006$ | $0.139 \pm 0.028$ |
| ✗ | ✓ | ✓ | $0.412 \pm 0.005$ | $0.118 \pm 0.077$ |
| **Ablation Study Over Cross-Validation Fitness** | | | | |
| ✓ | ✓ | ✗ | $0.356 \pm 0.009$ | $0.163 \pm 0.019$ |
| ✓ | ✗ | ✗ | $1.140 \pm 0.147$ | $0.0003 \pm 0.0002$ |

The top half of Table 1 shows how POPDESCENT's randomization (NN weights, learning, and regularization rates) lowers test loss by 25%. Adding noise and choosing models that perform well on cross-validation loss helps the models explore more space while selecting models that prevent overfitting, as see with a lower test loss. The bottom half shows how deciding between training or cross-validation loss as the fitness function acts as a substantial heuristic when minimizing test loss, genetically "forcing" a model without regularization to still achieve decent test loss. We present the most pronounced differences in Table 1 to best highlight POPDESCENT's features.

### 3.4  HYPERPARAMETER SENSITIVITY

In this section, we show that varying local search parameters affect ESGD (a state-of-the-art memetic algorithm) more than POPDESCENT on the CIFAR-10 dataset. We run each algorithm with a constant seed and constant hyperparameters except one (either learning rate or the number of iterations). One iteration defines one local and global update together. A gradient update is taken each time before performing a mutation. POPDESCENT defaults to a batch size of 64, a learning rate of 0.001 with Adam, and 30 iterations for the FMNIST benchmark. ESGD defaults to a batch size of 8, a learning rate of 0.01 with SGD, and 3 iterations for the FMNIST benchmark (POPDESCENT trains over 128 batches per iteration, ESGD over the whole training set).

Table 3 shows how changes in local training parameters affect ESGD's test loss results more than POPDESCENT's in Table 2 (almost 275% higher standard deviation of results). POPDESCENT also has a much lower test loss across trials (avg. 19.2% lower). Complex memetic algorithms such as ESGD have a high number of adjustable hyperparameters, and their performance depends significantly on their specific values. As long as the parameters chosen are not extreme values, the specificity of POPDESCENT's hyperparameters is not particularly important.

Table 2: Population Descent training with variable local parameters

| Learning Rate | Iterations | Test Loss $\pm \sigma$ Across Trials | $\sigma$ as % of Test Loss |
|---|---|---|---|
| **All Hyperparameters Constant Except Learning Rate** | | | |
| **[0.01, 0.05, 0.001]** | 30 | $1.049 \pm 0.172$ | 16.35% |
| **All Hyperparameters Constant Except Total Iterations** | | | |
| 0.001 | **[10, 30, 50]** | $0.958 \pm 0.191$ | 19.94% |

Table 3: ESGD training with variable local parameters

| Learning Rate | Iterations | Test Loss $\pm \sigma$ Across Trials | $\sigma$ as % of Test Loss |
|---|---|---|---|
| **Everything Constant Except Training Learning Rate** | | | |
| **[0.01, 0.05, 0.001]** | 3 | $1.325 \pm 0.582$ | 43.95% |
| **Everything Constant Except Total Iterations** | | | |
| 0.001 | **[1, 3, 5]** | $1.159 \pm 0.455$ | 39.22% |

Another important note is how we do not need to tune POPDESCENT over different problems, while still yielding the best model. All tests for POPDESCENT across this entire paper (except the ablation tests) use the same population size (5) and randomization scheme (same Gaussian noise distributions) for the global step, and the same default learning rate (0.001), regularization rate (0.001), and batch size (64) for the local step (except when they are changed for this experiment).

**Discussion on learning rate schedules.** Learning rate schedules (adjusting the learning rate over the number of gradient steps taken) are one of the most common way to "actively" adjust hyperparameters during training. However, most schedules are only a function of the number of gradient steps taken, which are only a prediction of training rather than analyzing real time how a model is performing like POPDESCENT does. Specifially, Table 3 shows how non-dynamic optimization algorithms (most existing methods) rely on problem-specific hyperparameters to be pre-determined. Modifications to the idea of learning rate schedules do exist, in order to pay attention to a model's progress Wu et al. (2019), though they are very complex and have many hyperparameters, running into sensitivity issues like ESGD.

## 4 RELATED WORKS

**Gradient-Based Optimizers.** Stochastic Gradient Descent (SGD) offers quick convergence on complex loss spaces Kleinberg et al. (2018). As an improvement to SGD, momentum-based optimizers like Adam Goh (2017); Zeiler (2012) better traverse loss functions via utilizing momentum with the learning rate to more quickly escape plateaus or slow down learning as to not skip a minima. Adam's weight decay term also limits exploding gradients, and acts as a regularizer preventing overfitting. Other options like the newer Sharpness-Awareness Minimization (SAM) or Shampoo optimizer, which use "preconditioning matrices," promise even faster convergence Gupta et al. (2018). POPDESCENT tunes hyperparameters for *any* local optimizer, including SGD, Adam, SAM, or Shampoo. Hence, such works are orthogonal.

**Existing Tuning Frameworks.** Grid search is the most commonly used method for searching the hyperparameter space due to its simplicity and generally acceptable performance for NN training. Essentially, it is an exhaustive search of the Cartesian product of hyperparameters. It takes exponentially more gradient steps to train over more parameter options, making grid search a more inefficient, brute-force way of tuning. Popular hyperparameter tuning frameworks like KerasTuner (KT) and Optuna employ more efficient alternatives to grid search Rogachev & Melikhova (2020); Akiba et al. (2019). This includes bayesian search (uses Gaussian probabilities to check the "best" combination) Mockus (1989), random search (randomly samples the search space) Bergstra et al.

(2011), or hyperband tuning (a variation of random search that chooses better individuals after half of the iterations) Rogachev & Melikhova (2020). They can sample different batches, batch sizes, learning/regularization rates, and even NN layers/units in order to find the best architecture for the task at hand. They often find a good set of hyperparameters within a constant amount of time as opposed to grid search's brute force method Liashchynskyi & Liashchynskyi (2019). However, these methods do not allow for dynamic hyperparameter optimization; each run is independent of progress made in previous runs, and most algorithms simply return hyperparameters that the model should be initialized with during training. Performance deteriorates during later training epochs, and the risk of overfitting is higher due to not adjusting parameters. Models trained with these tuners explore less of the loss space compared to evolutionary algorithms that mutate NN weights as well.

One common approach for adding dynamicity to hyperparameters is the use of schedules Li & Arora (2019). Learning rate schedules, for example, are often defined by three to five parameters that transition during training, and have been proposed to improve convergence speed Dauphin et al. (2015); Darken et al. (1992). These approaches are fundamentally limited as they are based on predictions about training, rather than a model's actual loss. Multiple works also explore cyclical, cosine-based, random-restart schedules adjusting the learning rate at every epoch Wu et al. (2019). They introduce extra hyperparameters that need to be tuned, causing many researchers to instead use static schedules.

**Meta-Learning.** Meta-learning is another topic that aims to find generalized models and hyperparameters that can improve the speed of convergence on *future* tasks. They therefore attack a different problem formulation than that of hyperparameter search, and have their limitations Gad (2022); Patacchiola et al. (2023). The body of work on hyper-parameter tuning, and therefore POPDESCENT, remains relevant even with the existence of meta-learning.

**Memetic Algorithms.** Memetic algorithms take advantage of both global and local learning, and are increasingly being used for supervised learning benchmarks D'Angelo & Palmieri (2021); Moscato (1999); Borna & Hashemi (2014); Xue et al. (2021); Yuenyong & Nishihara (2014). Evolutionary stochastic gradient descent (ESGD) Cui et al. (2018) is the state-of-the-art memetic algorithm that utilizes Gaussian mutations for model parameters using an $m$-elitist average strategy to choose the best models after randomization, and SGD optimization for local search. Performing well on CIFAR-10 classification tests, ESGD is a prime example of how adding stochastic noise benefits a strong local optimizer, because models explore more of the loss function. Nonetheless, memetic algorithms are not currently used for hyperparameter tuning. Also, state-of-the-art memetic algorithms like ESGD suffer from having an extensive amount of training hyperparameters, both global (ie. mutation strength, population size, etc.) and local (ie. batch size, learning rate, etc.). Thus, motivated by how stochastic noise can force models to explore more space, and how population-based methods can be used to globally optimize models, POPDESCENT instead investigates how it is possible use the evolutionary step in the memetic framework to tune local search hyperparameters much more dynamically than existing tuners or schedules. With a straightforward randomization scheme, our memetic framework is much simpler than existing algorithms, allowing higher efficiency during training. The simplicity also ensures our algorithm does not have to be tuned nearly as much for each specific problem. POPDESCENT is a new application of memetic algorithms specifically for hyperparameter tuning, so existing tuning frameworks are a more suitable comparison to POPDESCENT than memetic algorithms.

## 5 CONCLUSION

In this paper, we propose POPDESCENT, a memetic algorithm that acts as a hyperparameter tuning framework using a simple population-based evolution methodology. Our tuning framework helps local search methods explore more space on the loss function. In turn, it more effectively traverses a non-convex search-space compared to methods relying only on static momentum terms or schedules. POPDESCENT performs better than existing tuning frameworks which do not adapt to a model's current progress. Four extensive experiments over common supervised learning benchmarks FMNIST and CIFAR-10 show the effectiveness of POPDESCENT.

## 6 REPRODUCIBILITY STATEMENT

We take many efforts to make sure that our experiments can be reevaluated effectively:

- We use the number of gradient steps as our metric of "time", so that these values remain independent of the computational hardware available
- We always seed every experiment taken, and those seeds are available in our source-code.
- We provide versioned source code with specific commits referenced for each test taken, and provide a README of instructions to follow to replicate our results
- We provide our reference anonymized implementation of POPDESCENT and supplementary material at `https://github.com/anonymous112234/anonymousPopulationDescent.git`
- We provide a flake.nix file which exactly pins the versions of all the packages used in our tests

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
