# OpenReview forum: "POPULATION DESCENT: A NATURAL-SELECTION BASED HYPER-PARAMETER TUNING FRAMEWORK"
_ICLR.cc/2024/Conference — Submitted to ICLR 2024_

### Official Review · Reviewer_VVKB · 2023-10-13

**Soundness:** 2 fair
**Presentation:** 2 fair
**Contribution:** 2 fair
**Rating:** 3
**Confidence:** 2

**Summary:**

This paper propose a new algorithm combined with a fitness-based randomization scheme.

**Strengths:**

The algorithm is descried very detailedly.

**Weaknesses:**

1. The language used in this paper is not good. I recommend the author to use large language models (e.g. ChatGPT) to go through your work.

2. Too many tables, algorithms, and subjective comments in the paper. You should use more rigorous statements.

3. Based on the current version. I think the paper is more suitable for evolution journals like TEVC/Soft computation.

4. Too many irrelated sentences in the introduction. Everyone knows the property of global opt... Should make it more compact.

5. Fmnist and cifar10 are just too simple. Since it is a algorithm-like paper. Just doing such weak experiments are not enough.

**Questions:**

See weakness.

---

> ### Author Response · Authors · 2023-11-14
> **Response to Reviewer VVKB**
>
> Dear reviewer, we greatly appreciate all of the feedback to improve our paper. We will carefully incorporate the recommendations in the revised paper. In the following, your comments are first stated and then followed by our point-by-point responses.
>
> Comment: “The language used in this paper is not good. I recommend the author to use large language models (e.g. ChatGPT) to go through your work.”
> Extra proof-reading is a priority, and we will make sure to address as much of the poor language in the paper as possible.
>
> Comment: “Too many tables, algorithms, and subjective comments in the paper. You should use more rigorous statements.”
> Thank you for the suggestion. Currently, we are improving the clarity and conciseness of our tables. As for the point on subjective comments, we will try our best to replace them. If you could point out specific areas in need of more rigor, it would be very helpful to us.
>
> Comment: “Based on the current version. I think the paper is more suitable for evolution journals like TEVC/Soft computation.”
> We appreciate the specific recommendations. Even though our method is an evolutionary one, we do believe that this work has direct implications for training neural networks which should fall well within the scope of machine learning conferences like ICLR. We hope that making the method’s motivations clearer would convince the reviewers that this paper is suitable within ICLR’s domain.
>
> Comment: “Too many irrelated sentences in the introduction. Everyone knows the property of global opt... Should make it more compact.”
> We are currently rewriting the introduction to be more concise and better explain our contributions.
>
> Comment: “Fmnist and cifar10 are just too simple. Since it is a algorithm-like paper. Just doing such weak experiments are not enough.”
> We are expanding our experiments to include additional dataset benchmarks such as the Penn Treebank, and comparing against other state-of-the-art methods, including Sklearn’s Hyperopt and IRACE. We do use other tests to demonstrate PopDescent’s flexibility to not be extremely problem-specific; in Experiment 3.4, we show that PopDescent achieves quality results without parameter tuning, demonstrating the effectiveness of its 'active' tuning compared to other methods.

---

> > ### Comment · Reviewer_VVKB · 2023-11-23
> > **reply**
> >
> > I have gone through author's reply and other reviews's advice. I have no further questions.

---

### Official Review · Reviewer_zXbc · 2023-10-28

**Soundness:** 2 fair
**Presentation:** 2 fair
**Contribution:** 1 poor
**Rating:** 3
**Confidence:** 4

**Summary:**

This paper introduced a new simple evolutionary algorithm to support hyper-parameter tuning while training deep neural networks. Preliminary experiment results show that the new algorithm may be useful to some extent in practice.

**Strengths:**

Hyper-parameter tuning is important for many deep learning systems and many real-world applications. It is important to develop effective and efficient hyper-parameter tuning techniques. This paper presented a new attempt along this research direction.

**Weaknesses:**

This paper does not have strong technical novelty. The literature review was brief and did not cover many advanced tools and methods for hyper-parameter tuning or meta-learning in general. For example, the IRACE package is getting increasingly popular for hyper-parameter tuning. It remains largely questionable why it is necessary to develop a new evolutionary algorithm for hyper-parameter tuning, instead of using existing tools or technologies.

The design of the new evolutionary algorithm lacks technical novelty. It is common to use normalized fitness for individual selection in many evolutionary algorithms. It is also common to replace the worst individuals with mutated individuals. Controlling the randomness in mutation based on the performance/fitness of each individual is not new either. Furthermore, according to Algorithm 1, all individuals in the population need to be trained separately in each generation. This is computationally expensive and may not be as efficient as other gradient-based meta-learning techniques that can also fine-tune some hyper-parameters.

Besides the major concern on the technical novelty, the experimental evaluation is not sufficiently strong. Given the ever-expanding literature on hyper-parameter tuning techniques, the competing methods examined in the experiment appear to be quite limited, insufficient to show that the new algorithm can achieve state-of-the-art performance in both efficiency and effectiveness. Moreover, only two relatively simple benchmark datasets were utilized in the experiment. Results obtained on the two benchmark datasets cannot conclusively show the performance advantage of the new algorithm.

Some statements seem to be confusing. For example, I don't understand what the statement "actively choosing how much to explore the parameter and hyperparameter space" on page 2 means. It is also hard to understand the statement "struggle against fine-tuned local search solutions" on page 2.

The authors mentioned several limitations with the new algorithm in Subsection 2.3. It is not clear why they don't try to address these limitations, which appear to be closely relevant to the practical usefulness of the new algorithm and cannot be simply declared as future works.

Typos and grammatical errors can be spotted frequently throughout the paper. The authors are highly recommended to conduct more rounds of proof-reading to significantly improve the presentation quality and clarity of this paper.

**Questions:**

Why is it necessary to develop a new evolutionary algorithm for hyper-parameter tuning, instead of using existing tools or technologies?

Compared to other hyper-parameter tuning and meta-learning techniques, how efficient is the newly proposed evolutionary algorithm and why?

Can the new algorithm achieve state-of-the-art performance in both efficiency and effectiveness and why?

Why didn't the authors try to address the limitations discussed in Subsection 2.3?

---

> ### Author Response · Authors · 2023-11-14
> **Response to Reviewer zXbc**
>
> Dear reviewer, thank you for your constructive comments, they are very helpful for us in improving our paper. In the following, your comments are first stated and then followed by our point-by-point responses.
>
> Comment: “Can the new algorithm achieve state-of-the-art performance in both efficiency and effectiveness and why?”
> Recognizing that our current experimental section may not be fully convincing in proving our claims, we plan to enhance it by adding more dataset benchmarks such as the Penn Treebank and comparing with other state-of-the-art methods like Sklearn’s Hyperopt and IRACE.
>
> The reason we believe PopDescent is more efficient and effective is demonstrated via comparisons with existing hyper-parameter tuning techniques. We show that PopDescent can achieve lower losses in fewer gradient steps (a metric of computational time that is invariant to the computational power available) on FMNIST and CIFAR-10. In Table 1, PopDescent achieved a 15.2% lower test loss on the FMNIST dataset with regularization compared to the next best method. Similarly, on the CIFAR-10 dataset with regularization, PopDescent achieved a 13.1% lower test loss compared to the next best method. In both cases, PopDescent used 31.6% and 34.4% fewer gradient steps, respectively.
>
> Comment: “The design of the new evolutionary algorithm lacks technical novelty.”
> We will reframe our contributions focusing on the novelty in designing a memetic algorithm specifically for hyper-parameter tuning. We will convey that the trade-offs we make for the design choices forming the whole algorithm are novel, rather than the individual pieces themselves. While features like using “normalized fitness”, using m-elitist selection, or controlling the “randomness in mutation” are not our creations, we combine them uniquely to introduce an algorithm that has not been created before. To our knowledge, PopDescent’s design has never been explored, as memetic algorithms are not designed to be insensitive to their own hyper-parameters. Experiment 3.4 substantiates that claim by showing ESGD’s sensitivity to its hyper-parameters while showing that PopDescent is resilient. Requiring re-tuning of parameters for each problem defeats the purpose of a hyper-parameter search method. To our knowledge, the design space of hyper-parameter tuning frameworks that perform constant model selection, randomization, and updates, is under-explored. As mentioned, other frameworks only partially test hyper-parameters over the first few iterations.
> Comment: “All individuals in the population need to be trained separately in each generation. This is computationally expensive and may not be as efficient as other gradient-based meta-learning techniques that can also fine-tune some hyper-parameters.”
> Meta-learning aims to find generalized models and hyper-parameters that can improve the speed of convergence of learning on future similar tasks. In some sense, they attack slightly different problem formulations than hyper-parameter search does and have their limitations (see [Massimiliano P. et al  arXiv:2306.13554]). As such, our body of work on hyper-parameter search is still relevant even with the existence of meta-learning. Our method therefore has the same motivations as other novel hyper-parameter-search techniques. Thus, we believe comparisons against such techniques are appropriate. If there is a method PopDescent should be compared against that we currently fail to address, we would be grateful for any suggestions.
> Existing hyper-parameter tuning frameworks also train individual models separately; we argue that parallelism provides an advantage for scaling training. PopDescent still uses fewer gradient steps to find models reaching similar/better losses (Table 1).
> Comment: “The experimental evaluation is not sufficiently strong.”
> See response 1.
>
> Comment: “It is not clear why they don't try to address these limitations.” (referring to section 2.3)
> We will revise the section addressing the limitations. Specifically, we gave pathological examples where PopDescent fails. We will address them by adding an excerpt mentioning that these cases are never observed in our experiments; we hypothesize that standard machine learning problems do not form such pathological cases.
>
> Comment: “I don't understand what the statement "actively choosing how much to explore the parameter and hyper-parameter space" on page 2 means.”
> This statement refers to how our algorithm controls the randomness in mutation to randomize worse-performing individuals more. The term “active” refers to how PopDescent makes this adjustment in training every iteration. We will make the phrasing clearer in the paper.
>
> Comment: It is also hard to understand the statement "struggle against fine-tuned local search solutions" on page 2.”
> This statement refers to the sensitivity that the existing memetic algorithms have towards their own hyper-parameters. We will reword the statement for clarity.

---

> > ### Comment · Reviewer_zXbc · 2023-11-22
> > **Thank you the authors for their response**
> >
> > I would like to thanks the authors for their response to my questions.
> >
> > I found that the argument of a unique combination of existing techniques is not sufficiently convincing. There are many ways to combine existing techniques, which can be claimed as "unique". However, the deep theoretical reasons for such a combination to be particularly effective is not explored to a full extent. IMHO, this aspect should be improved to strengthen the overall technical contribution of this paper. In the same vein, limited experiment results cannot completely prove that PopDescent is highly resilient to hyper-parameter variations under any specific conditions. More theoretical analysis is necessary to consolidate this claimed contribution.

---

### Official Review · Reviewer_TkFw · 2023-10-31

**Soundness:** 3 good
**Presentation:** 2 fair
**Contribution:** 3 good
**Rating:** 5
**Confidence:** 5

**Summary:**

The paper proposes population descent, or PopDescent - a memetic algorithm that combines local gradient-based search with a global population-based search. Local search is applied to traverse the parameter space (per individual in a population), and global search is applied to traverse hyper-parameters. The method is deliberately simple, and designed to not be sensitive to its own hyperparameters. Experiments demonstrate that the proposed method effectively optimizes both the problem and the local algorithm search parameters (regularisation, learning rate).

**Strengths:**

In the gradient-dominated field of neural network optimisation, it is refreshing to see an approach that attempts to bring global-search algorithms to the table without severe efficiency trade-offs.

**Originality:** The proposed method seems reasonably novel, although I would have appreciated a more critical comparison to other memetic algorithms of similar kind.

**Quality and clarity:** The paper is easy to follow, pseudo code is provided for the proposed algorithms. Authors also provide an anonymised link to their code, which is a big plus.

**Significance:** On two benchmarks (FMINST and CIFAR-10), the proposed method is shown to outperform both random parameter search and a competing memetic algorithm.

**Weaknesses:**

**Literature review:** The authors propose a new memetic algorithm, but fail to sufficiently discuss existing state-of-the-art memetic algorithms in their literature review. ESGD is briefly mentioned in the “Benchmarks” section, but its workings are not described or critically compared to the proposed approach. Evolutionary/population-based algorithms are plenty, and without a critical discussion of existing methods, it is hard to evaluate the originality of the proposed method. I am also not certain why authors decided to move related work discussion to just before the conclusions - this does not make for a good narrative structure, and should be moved to the beginning of the paper.

**Experimentation:** In the experiments, all methods employ Adam except for ESGD. This seems like an unfair comparison: perhaps the superiority of the proposed method is due to the superior performance of Adam as compared to SGD? Adam is known to converge faster than SGD, which might also explain why the proposed method converged quicker than ESGD.

Authors list the total number of parameters, but do not specify the architectures of the CNNs use (how many channels p/layer etc.).

**Questions:**

Authors use cross-validation error to perform genetic evolution of the hyperparameters. Isn’t this going to leak information about the test set? The final errors reported - are they calculated on some hold-out set that is not seen during the evolutionary process?

Formatting issues:
1. Citations are not enclosed in parenthesis - for example, “…Large Language Models Cheng et al. (2023)” - should rather be “…Large Language Models (Cheng et al., 2023)”
2. Acronyms: there is no need to capitalize every word that is going to be abbreviated. I.e., instead of “Neural Networks (NNs)” one can simply write “neural networks (NNs)”.
3. “when the magnitude is at 0, None” -> when the magnitude is at 0, none
4. “as see with a lower” -> as seen with a lower
5. “One iteration defines one local and global update together. gradient updates the algorithm takes before performing a mutation.” - the two sentences seem malformed.

---

> ### Author Response · Authors · 2023-11-14
> **Response to Reviewer TkFw**
>
> Dear reviewer, thank you for your insightful questions and carefully reviewed comments, they are exceedingly helpful for us to improve our paper. We will carefully incorporate them in the revised paper. In the following, your comments are first stated and then followed by our point-by-point responses.
>
> Comment: “The authors propose a new memetic algorithm, but fail to sufficiently discuss existing state-of-the-art memetic algorithms in their literature review.”
> Yes, our description of memetic algorithms is not as in-depth as our discussion on Tuning Frameworks. However, our decision is based on applying a memetic framework to hyper-parameter tuning, making PopDescent unique. We were initially “motivated by the difficulties in using a memetic framework for fine-tuning hyper-parameters.” Thus, we compared PopDescent to ESGD, but the method more directly competes against other state-of-the-art tuning frameworks like KerasTuner. There aren’t many other memetic algorithms that are designed for this task. However, we will compare our results with more existing memetic algorithms and more tuning frameworks which should strengthen our claims.
>
> Comment: “I am also not certain why authors decided to move related work discussion to just before the conclusions.”
> Thank you for this suggestion, we are moving the related works to after the introduction for a better narrative structure. Initially, we followed the structural recommendations from: https://www.microsoft.com/en-us/research/academic-program/write-great-research-paper/. However, much of the motivation of our method comes from its relation to existing works, so we believe the literature review section should take higher precedence.
>
> Comment: “In the experiments, all methods employ Adam except for ESGD. This seems like an unfair comparison: perhaps the superiority of the proposed method is due to the superior performance of Adam as compared to SGD? Adam is known to converge faster than SGD, which might also explain why the proposed method converged quicker than ESGD.”
> Thank you for pointing this out. Our statement in Section 3.1 “All algorithms use the Adam optimizer for local search, except for ESGD, which uses SGD”, is confusing. The original work makes use of SGD as its local optimizer. However, in our benchmark evaluations, we used the optimizer that ESGD performs best with: SGD for FMNIST, and Adam for CIFAR-10. This choice is apparent in our code, but we failed to mention it in the paper, our revision will reflect this choice. We tried to make the comparison with ESGD as fair as possible. If there are any more suggestions on how we could do so, we would appreciate them greatly. We will also add additional experiments bolstering our claims, such as adding the Penn Treebank dataset and comparing PopDescent against other state-of-the-art methods like Sklearn’s Hyperopt, and IRACE.
>
> Comment: “Authors list the total number of parameters, but do not specify the architectures of the CNNs use (how many channels p/layer etc.).”
> We describe the overall makeup of the models used, such as: “An identical Convolutional Neural Net with 4,575,242 parameters, consisting of three convolutional layers and two fully connected layers.” However, the exact details of each layer will be added to our revised supplementary material.
>
> Comment: “The final errors reported - are they calculated on some hold-out set that is not seen during the evolutionary process?”
> All final errors reported (except for ESGD, which we use as instructed in the provided open-source implementation), are calculated on a hold-out set that is not seen during the evolutionary process. For all methods (except for ESGD), we split the test-set into a “cross-validation set” and “test-set.” The evolutionary processes rely on the “cross-validation set” and the final errors are calculated on the “test-set.”
>
> In the paper, we describe it as follows: “These individuals are maximized on batches from a test distribution that remains unseen during the procedure. Since the Test distribution must remain unseen, we are forced to make use of available proxy data in the form of a Training distribution and a CrossValidation distribution.” We have revised our statement, renaming this unseen dataset as a “hold-out” set in the paper for clarity.
>
> Formatting issues and further proofreading rounds will be applied for the final revision before the rebuttal period ends.

---

### Official Review · Reviewer_nmn7 · 2023-11-05

**Soundness:** 2 fair
**Presentation:** 1 poor
**Contribution:** 1 poor
**Rating:** 3
**Confidence:** 4

**Summary:**

This paper proposes a memetic algorithm, Population Descent, which combines the benefits of gradient descent and black-box optimization methods. POPDESCENT is based on population-based evolution, helping to explore more space in the loss function and performing better than existing frameworks. Experiments on FMNIST and CIFAR-10 datasets demonstrate its effectiveness.

**Strengths:**

1. The considered problem is important.
2. The idea of combining first-order optimizer and black-box optimizer is sound.

**Weaknesses:**

1. The problem formulation in this paper is rather unconventional. To enhance the overall coherence of the paper, I recommend commencing with an overview of Black-box optimization and leveraging the context of evolutionary algorithms to guide the logical progression.
2. The introduction of the problem is overly simplistic and fails to provide an in-depth explanation of non-convex optimization and saddle points. Additionally, it lacks essential references on these topics, which are crucial for a comprehensive understanding.
3. The novelty of the proposed method appears to be limited in comparison to existing approaches.
4. The experimental section is notably inadequate in terms of datasets and compared methods. It is essential to incorporate a comparison with advanced optimizers, such as the Sharpness-aware optimizer, to provide a more comprehensive evaluation and gauge the effectiveness of the proposed method against state-of-the-art techniques.
5.  Furthermore, the experimental results exhibit mediocre performance and lack clarity in demonstrating significant effects.

**Questions:**

See Weaknesses

---

> ### Author Response · Authors · 2023-11-14
> **Response to Reviewer nmn7**
>
> Dear reviewer, thank you for your constructive comments, they are exceedingly helpful for us to improve our paper. We have carefully incorporated them in the revised paper. In the following, your comments are first stated and then followed by our point-by-point responses.
>
> Comment: “The problem formulation in this paper is rather unconventional. To enhance the overall coherence of the paper, I recommend commencing with an overview of Black-box optimization and leveraging the context of evolutionary algorithms to guide the logical progression.”
> We appreciate this thoughtful suggestion. We will enhance the logical progression of the introduction by providing an overview of Black-box optimization and leveraging the context of evolutionary algorithms for PopDescent.
>
> Comment: “The introduction of the problem is overly simplistic and fails to provide an in-depth explanation of non-convex optimization and saddle points.”
> While revamping our introduction for the problem formulation, we are bolstering the explanation of the problem itself, especially regarding non-convex optimization and saddle points. We are also currently adding more references for these topics to enhance the introduction.
>
> Comment: “The novelty of the proposed method appears to be limited in comparison to existing approaches.”
> We will reframe our contributions focusing on the novelty in designing a memetic algorithm specifically for hyper-parameter tuning. We will convey that the trade-offs we make for the design choices forming the whole algorithm are novel, rather than the individual pieces themselves. While features like using “normalized fitness”, using m-elitist selection, or controlling the “randomness in mutation” are not our creations, we combine them uniquely to introduce an algorithm that has not been created before. To the best of our knowledge, we are the only algorithm to use the mentioned features in a combination that yields a much simpler tuning process: in the paper, we mention how other state-of-the-art memetic algorithms’ “genetic schemes are complex and thus difficult to implement.” PopDescent does not need fine-tuning of any parameters, however, as shown by Experiment 3.4. Also, PopDescent is, to the best of our knowledge, the most “active” hyper-parameter tuning framework, meaning it makes the most adjustments during training, instead of beforehand. As mentioned, other frameworks only partially test hyper-parameters over the first few iterations and lack precision during later phases of training, or ignore efficiency and train all possible outcomes. Similarly, PopDescent improves upon commonly used hyper-parameter “schedules,” which do not adjust rates based on training progress. We appreciate the feedback on presenting novelty, and we are currently rewording our contributions to highlight how PopDescent is 1) a new memetic algorithm given its unique combination of features, 2) uniquely applied to hyper-parameter tuning, and 3) a more “active” tuner and thus better performing/more efficient than existing tuning frameworks.
>
> Comment: “The experimental section is notably inadequate in terms of datasets and compared methods. It is essential to incorporate a comparison with advanced optimizers, such as the Sharpness-aware optimizer, to provide a more comprehensive evaluation and gauge the effectiveness of the proposed method against state-of-the-art techniques.”
> We are working to add both tests on more dataset benchmarks like the Penn Treebank dataset, and against other state-of-the-art methods like Sklearn’s Hyperopt, and IRACE. The Sharpness-aware optimizer is designed as a competitor to optimizers like SGD, Adam, or Shampoo. However, as we state in the paper’s related works, “PopDescent relies on efficient local optimizers.” PopDescent is more directly comparable to hyper-parameter tuning algorithms instead. The Sharpness-Aware optimizer can replace Adam in any of our tests, as our paper does not rely on which specific local optimizer is used, and tunes hyper-parameters regardless.
>
> Comment: “Furthermore, the experimental results exhibit mediocre performance and lack clarity in demonstrating significant effects.”
> We are reformatting our tables to better demonstrate the significance of our experiments. In Table 1 benchmarks, PopDescent achieved a 15.2% lower test loss on the FMNIST dataset with regularization compared to the next best method. Similarly, on the CIFAR-10 dataset with regularization, PopDescent achieved a 13.1% lower test loss compared to the next best method. In both cases, PopDescent used 31.6% and 34.4% fewer gradient steps, respectively. PopDescent also converges quicker, as shown in Figure 1. The ablation study shows how PopDescent’s randomization scheme helped an identical model achieve 16.2% lower test loss on the FMNIST dataset, compared to not using PopDescent’s randomization. We would love to know if there are any tests in mind that clearly prove/disprove our hypothesis about PopDescent's capabilities.

---

### Meta-Review · Area_Chair_6Cr8 · 2023-12-04

**Metareview:**

This paper received scores of 3,3,3,5, and the reviewers provided a long list of weaknesses. The authors rebutted, but the reviewers maintained their rejection rankings. In the discussion phase, also none of the reviewers argued in favor of the paper. Hyperparameter optimization methods with new state-of-the-art performance are clearly important to ICLR, but that state-of-the-art performance needs to be demonstrated on a diverse set of current HPO benchmarks and include current state-of-the-art hyperparameter optimization methods. I encourage the authors to look through the last few NeurIPS, ICML and ICLR proceedings, searching for "hyperparameter optimization" to find much stronger baselines than they used.
I recommend rejection and resubmission to a different venue; I agree with reviewer VVKB that evolutionary computation venues would likely be a better fit for the current focus of the work.

**Justification For Why Not Higher Score:**

Nobody is in favor of that.

**Justification For Why Not Lower Score:**

N/A

---

### Decision · Program_Chairs · 2024-01-16

Reject